# A New Improvement Proposal to Estimate Regional Input–Output Structure Using the 2D-LQ Approach

Rubén Martínez-Alpañez [1], José Daniel Buendía-Azorín [1]  and María del Mar Sánchez-de-la-Vega [2,*] 

[1] Department of Applied Economics, University of Murcia, 30100 Murcia, Spain
[2] Department of Quantitative Methods for Economy and Business, University of Murcia, 30100 Murcia, Spain
[*] Correspondence: marvega@um.es

**Abstract:** The use of location quotients for the estimation of regional input–output tables has been found to be a useful and efficient tool to estimate intraregional production coefficients and multipliers. This paper considers some regionalisation methodologies based on location quotients for the estimation of input–output tables—some of which have hitherto not been analysed at the regional level—and studies which one provides the best estimation (best goodness of fit). We focus the analysis mainly on the accuracy of Flegg's location quotient (FLQ) and two-dimensional location quotient (2D-LQ). The analysis makes use of the multiregional input–output table for Korea for the year 2015 to evaluate the accuracy of the 2D-LQ method against FLQ. A novel proposal for the determination of the parameters corresponding to the 2D-LQ method is presented. This proposal is evaluated in Korean regions and is also applied to Spanish regions. The results obtained from the research conclude the general superiority of the 2D-LQ method, thus corroborating the results of other studies at the national level as well as the validity of our proposal.

**Keywords:** location quotients; FLQ; 2D-LQ; regional input–output models; interregional trade flows; regression analysis



## 1. Introduction

The use of location quotients (LQ) allows the input–output analysis to be extended to regional territories without an input–output framework, estimating the regional input–output table in the absence of survey data. However, the result obtained must be reviewed by the analyst and refined in order to obtain the best possible approximation to the interindustrial economic reality of the estimated territory in question (Flegg and Webber 2000).

Either directly recognising its better closeness of fit compared with other techniques (Jahn 2017; Lampiris et al. 2019), or indirectly using it as a comparative reference, regardless of its determination as the best estimator (Kowalewski 2015; Lamonica and Chelli 2018; Zhao and Choi 2015), there is a significant amount of work, for example, (Flegg et al. 2016; Lampiris et al. 2019; Romero et al. 2012) that make explicit mention in the comparison of methodologies to the higher precision in the estimation of Flegg's location quotients, FLQ (Flegg et al. 1995; Flegg and Webber 1997).

The FLQ methodologies and their augmented version AFLQ (Flegg and Webber 2000) will entail a highly significant dependence on the value given to the parameter δ (Flegg and Tohmo 2019; Kowalewski 2015; Lamonica and Chelli 2018; Lampiris et al. 2019), which is very costly to determine. It is worth noting that, in practice, this augmented alternative AFLQ does not generally perform better than its simple version FLQ (Bonfiglio 2009), hence it is still the FLQ alternative that continues to receive the most attention as the most suitable for undertaking regionalisation processes.

On the other hand, the two-dimensional location quotients methodology, 2D-LQ, (Pereira-López et al. 2020, 2021) has recently been presented and tested on the Input Output

Tables (IOTs) of six European economies (Pereira-López et al. 2020), derived from the input–output tables of the Euro Area 17 ( EA17 IO matrix) available in Eurostat. The result in both coefficients and multipliers, according to the authors, offers better approximations to the true parameters than the FLQ and AFLQ methodologies. Specifically, a comparison is made between the AFLQ methodology using the optimal δ and the 2D-LQ methodology, with the 2D-LQ estimation showing higher accuracy.

However, Flegg et al. (2021) argue in relation to the evaluation of the 2D-LQ methodology that, while acknowledging its adequate theoretical foundation, there are limitations related both to the determination of the parameters associated with the rectifications and to the national level on which it has been tested, stating that it would be more appropriate to test its accuracy at subnational levels given the more than foreseeable greater dependence of interregional trade on subnational territories.

In this context, the aim of this paper is to determine which LQ methodology provides the most accurate estimations of regional input–output tables, as well as to provide some new proposals for the implementation of such methodologies. While the FLQ range of methodologies has been evaluated at the regional level, the recent 2D-LQ methodology has not been studied in this framework. Moreover, there have hitherto not been proposals on the estimation of the parameters involved in 2D-LQ. Thus, this paper's main contributions consist, on the one hand, in providing a proposal for estimating these parameters, and on the other hand, in studying which is the best LQ regionalisation methodology in different scenarios, including our proposal in the comparison.

Based on Korea's 2015 multiregional input–output matrix, this paper evaluates different methodologies based on location quotients, paying special attention to the FLQ and 2D-LQ methodologies. In this context, we propose a novel procedure for estimating the parameters associated with both methodologies. The rest of the paper is structured as follows. Section 2 presents and describes the different location coefficient methods used in this work. Section 3 presents the data source and the statistics used to measure goodness-of-fit. Section 4 presents the comparative results obtained from the FLQ and 2D-LQ methodologies applied to the Korean multiregional table for the year 2015 and includes the proposal we made for the estimation of the unknown parameters for the 2D-LQ methodology. We then analyse and evaluate the results obtained for both the Korean and Spanish regions. Finally, the main conclusions drawn are summarised.

## 2. Methodological Review on Location Quotients

Since the publication of the original work by W. Isard in 1951, the use of location quotients and their variants in regional input–output analysis has been a constant over time. The methodological contributions that have been incorporated into the original approach from the perspective of the methodologies implemented and tested in this paper are presented in detail below.

Following the generalised nomenclature, let $x_i^r$ and $x^r$ be the total output of sector $i$ in region $r$ and the total output of region r, respectively, and let $x_i^n$ and $x^n$ be the respective totals referring to the national level; the location quotient for sector $I$ in region $r$ can be defined as:

$$SLQ_i = LQ_i^r = \frac{\frac{x_i^r}{x^r}}{\frac{x_i^n}{x^n}}. \tag{1}$$

It is most useful to present this quotient by performing a simple algebraic transformation in the following form:

$$SLQ_i = LQ_i^r = \frac{\frac{x_i^r}{x_i^n}}{\frac{x^r}{x^n}}. \tag{2}$$

where the numerator presents the share of total national output of product $i$ produced in region $r$, and the denominator represents the share of total regional output in the national total.

Note that, in practice, given that we are trying to establish regional shares of national totals, it is common, given the availability of information, to use data other than national production. Thus, employment data are commonly used (Kowalewski 2015; Miller and Blair 2009; Sargento et al. 2012), although others such as sectoral value added, income, and others showing such proportionality can be used (Flegg et al. 2014; Jahn 2017).

The regional coefficient $a_{ij}^{rr}$, which is the difference between the regional technical coefficient $a_{ij}^{r}$ and the regional import coefficient $a_{ij}^{sr}$, is derived from the adjustment by the location quotient of the national coefficient $a_{ij}^{n}$ for each industry. Thus:

$$a_{ij}^{rr} = \begin{cases} (SLQ_i^r)a_{ij}^n, & if \ SLQ_i^r < 1 \\ a_{ij}^n, & if \ SLQ_i^r \geq 1 \end{cases}. \tag{3}$$

The regional coefficient $a_{ij}^{rr}$ coincides with the national coefficient if the location quotient is greater than unity, due to the established assumption of coincidence of productive structure between the region and the higher national level. On the other hand, the coefficient is rectified when the location quotient is less than unity, on the understanding (not as in the previous case) that the difference derives from the existence of imports.

Based on the need for the location quotient to adequately include a measure of interregional trade, Round (1978) presents the semilogarithmic location quotient, which depends both on the regional size of the selling sectors and the buying sectors, as well as the size of the region itself. Thus:

$$RLQ_{ij} = \frac{SLQ_i}{\log_2(1 + SLQ_j)}. \tag{4}$$

A variant of the semilogarithmic location quotient (*RLQ*) seen above is the interindustry location quotient attributed to Charles Leven by Tiebout in 1966 (Schaffer and Chu 1969). This quotient seeks to make cell-by-cell adjustments rather than row-by-row adjustments to the matrix, considering the relative size of both the selling sectors, *i*, and the buying sectors, *j* (Ramos 1998).

The Cross-Industry Location Quotient, *CILQ*[1], is defined as follows:

$$CILQ_i = \frac{x_i^r / x_i^n}{x_j^r / x_j^n} \tag{5}$$

Then:

$$a_{ij}^{rr} = \begin{cases} (CILQ_{ij}^r)a_{ij}^n, & if \ CILQ_{ij}^r < 1 \\ a_{ij}^n, & if \ CILQ_{ij}^r \geq 1 \end{cases} \tag{6}$$

Comparing the relative sizes of sectors *i* and *j*, it is assumed that, if *CILQ* is less than unity, the relative size of sector *i* is smaller than the relative size of sector *j* in the region under analysis, so that it needs to import products to satisfy the demand of *j*.

As can be deduced (Miller and Blair 2009), $CILQ_{ij}^r = LQ_i^r / LQ_j^r$, so that the elements where *i* = *j* are equal to unity. In this case, a rectification is necessary (Flegg et al. 1995), completing the evaluation of the quotient for the determination of the coefficient $a_{ij}^{rr}$ in the following way:

$$\begin{aligned} a_{ij}^{rr} &= \begin{cases} (CILQ_{ij}^r)a_{ij}^n, & if \ CILQ_{ij}^r < 1 \\ a_{ij}^n, & if \ CILQ_{ij}^r \geq 1 \end{cases} \quad for \ i \neq j, \\ a_{ij}^{rr} &= \begin{cases} (SLQ_{ij}^r)a_{ij}^n, & if \ SLQ_i^r < 1 \\ a_{ij}^n, & if \ SLQ_i^r \geq 1 \end{cases} \quad for \ i = j. \end{aligned} \tag{7}$$

The *SLQ* and *CILQ* have certain limitations, such as the overestimation of intraregional trade, underestimating interregional trade or, for example, the fact that the production structure of a given territory has a higher or lower share of procurement relative to the national average (Flegg et al. 1995; McCann and Dewhurst 1998; Miller and Blair 2009). In

an attempt to improve them, *FLQ* (Flegg and Webber 1997) is implemented. It is defined as follows:

$$FLQ_{ij}^r = CILQ_{ij}^r(\lambda) \text{ where } \lambda = \left( log_2 \left( 1 + \frac{x^r}{x^n} \right) \right)^\delta, 0 \le \delta \le 1. \tag{8}$$

Then:

$$a_{ij}^{rr} = \begin{cases} (FLQ_{ij}^r)a_{ij}^n, & if\ FLQ_i^r < 1 \\ a_{ij}^n, & if\ FLQ_i^r \ge 1 \end{cases}. \tag{9}$$

In order to properly capture the possible regional specialisation that would lead a given region to be more specialised than indicated by the national coefficient, Flegg's methodological proposal was modified with a new proposal (Flegg and Webber 2000), the *AFLQ*, whose expression is:

$$AFLQ_{ij}^r = \begin{cases} \left[ log_2 \left( 1 + LQ_j^r \right) \right] (FLQ_{ij}^r) & if\ SLQ_j^r > 1 \\ FLQ_{ij}^r, & if\ SLQ_j^r \le 1 \end{cases}. \tag{10}$$

Additionally, in this way:

$$a_{ij}^{rr} = \begin{cases} (AFLQ_{ij}^r)a_{ij}^n, & if\ SLQ_j^r > 1 \\ (FLQ_{ij}^r)a_{ij}^n, & if\ SLQ_j^r \le 1 \end{cases}. \tag{11}$$

On the other hand, the two-dimensional location quotient (Pereira-López et al. 2020) is based on the premise that the adjustment needed in the regionalisation process for the cost structure of a given industry does not necessarily have to be related to the adjustment needed in the sales structure, allowing a different adjustment parameter to be chosen for each of the two cases.

The characteristic elements of the matrix of intermediate coefficients, $\widetilde{A}^r = \left( \widetilde{a}_{ij}^r \right)_{i,j=1,2,\ ...,\ m}$, are to be defined from the following expression:

$$\widetilde{A}^r = R(\alpha)A^n S(\beta). \tag{12}$$

where $A^n = \left( a_{ij}^n \right)_{i,j=1,2,\ ...,\ m}$ is the matrix of the national coefficients, and $R(\alpha)$ and $S(\beta)$ are diagonal matrices whose elements are null, except for those of the main diagonal, that is,

$$R(\alpha) = \begin{pmatrix} r_1(\alpha) & 0 & \cdots & 0 \\ 0 & r_2(\alpha) & 0 \cdots & 0 \\ \vdots & \vdots & \ddots & \vdots \\ 0 & 0 & & r_m(\alpha) \end{pmatrix} \text{ and } S(\beta) = \begin{pmatrix} s_1(\beta) & 0 & \cdots & 0 \\ 0 & s_2(\beta) & 0 \cdots & 0 \\ \vdots & \vdots & \ddots & \vdots \\ 0 & 0 & & s_m(\beta) \end{pmatrix}. \tag{13}$$

where:

$$r_i(\alpha) = (SLQ_i)^\alpha \quad i = 1, 2, \ldots, m \text{ and} \tag{14}$$

$$s_j(\beta) = \left( wx_j^r \right)^\beta \text{ with } wx_j^r = x_j^r / x_j^n \quad j = 1, 2, \ldots, m. \tag{15}$$

Therefore, from (12):

$$\widetilde{a}_{ij}^r = r_i(\alpha)a_{ij}^n s_j(\beta) \quad i, j = 1, 2, \ldots, m. \tag{16}$$

Thus, both regional specialisation and regional size are corrected by the values of the matrices $R(\alpha)$ and $S(\beta)$, respectively.

Depending on the value of the simple location coefficient *SLQ*, this methodology causes the elements of the regionalised matrix to take the following values

$$\tilde{a}_{ij}^{r} = \begin{cases} (SLQ_i^r)^\alpha a_{ij}^n \left(wx_j^r\right)^\beta & if \ SLQ_i^r \le 1 \\ \left[\frac{1}{2}tanh(SLQ_i^r - 1) + 1\right]^\alpha a_{ij}^n \left(wx_j^r\right)^\beta & if \ SLQ_i^r > 1 \end{cases} \tag{17}$$

where $tanh(SLQ_i^r - 1)$ is the hyperbolic tangent of $SLQ_i^r - 1$. The correction made through the hyperbolic tangent function allows the estimated regional coefficients to be 'slightly higher' than the corresponding national coefficients if $SLQ_i^r > 1$ (Pereira-López et al. 2020, p. 480).

As indicated above, the 2D-LQ methodology has been tested on the IOTs of six European economies, derived from the EA17 IO matrix available from Eurostat. The results in both coefficients and multipliers, according to the authors, provide better approximations of the true parameters than other more commonly used methodologies, namely: *FLQ* and *AFLQ*. Specifically, a comparison is made between the AFLQ methodology using the optimal δ and the 2D-LQ methodology, with the 2D-LQ estimation proving to be more accurate, except in one case. Furthermore, in a later work (Pereira-López et al. 2021), the authors confirm the superiority of 2D-LQ by comparing it with the rest of the quotients on the estimation of ten input–output tables corresponding to the years 2010 and 2015 for five European countries.

In this methodology, the α and β values that correct the parameters associated with the rectification of the national coefficients are decisive. Thus, the parameter α associated with the value of the *SLQ* in (14) corrects this value according to the degree of productive specialisation in the region. The correction associated with (17) shows whether the region is oriented towards imports of certain products (in the case of *SLQ* < 1) or whether the sector can be considered an exporter, based on covering the product needs of the region itself (in which case *SLQ* > 1). The value of α determines a higher or lower estimate of domestic intermediate consumption, depending on whether its value is above or below unity (Pereira-López et al. 2020). On the other hand, the parameter β associated with the columnar rectification of the national coefficients is related to the relative size of each industry, unlike the *FLQ* and *AFLQ* methodology.

In this way, the rectification made to the national coefficients is twofold, taking into account productive specialisation and making a smoothing (in this case following a tangential function and, in the case of *AFLQ*, semilogarithmic) according to the size of the sector to be applied when the simple location quotient shows values greater than unity.

In this context, the process of parameterisation of the α and β coefficients is not without complexity and a certain degree of difficulty that affects the measurement of the closeness of fit (precision) of the estimates. Thus, the authors confirm the lower sensitivity of the precision of the estimation with regard to changes in the parameters, compared with that related to the parameter associated with the Flegg quotients in their two versions.

This paper presents an empirical comparison between the above *LQ* methodologies of regional input–output table estimation. This comparison is focused on the following aspects:

- What the differences between the estimated and the real matrices are.
- Which method can provide the most accurate estimation.
- How the estimates vary in response to changes in the parameters.
- The precision achieved with the proposals presented in this paper.

To address the first question, the differences between the estimated and real matrices are measured by means of a goodness-of-fit statistic, which is performed for the different scenarios considered. The second point is analysed by comparing the deviations from the real values of the best estimates that the different methodologies can obtain, i.e., the values of the parameters involved in the design of the methodology that give rise to the most accurate estimation (optimal values) are considered in each case. For the third issue,

methodologies with different combinations of the associated parameters are compared. Finally, to address the last question, we calculate the deviations in respect to the real values of the estimations obtained when considering the parameters calculated using our proposal, and these are then compared with the results obtained in the previous analysed cases.

Korea's multiregional table for 2015 was used for the empirical study, as it is a recent database with a wide availability of survey data at the regional level. However, in the case of regions without a multiregional input–output framework, the application was carried out in the Spanish regions.

## 3. Data Source and Closeness of Fit

In order to properly test the estimated coefficients against the true (survey) coefficients, the homogeneous and uniform data for Korea from the multiregional input–output table for the year 2015 are used.

Korea's multiregional table for 2015 presents a $33 \times 33$ product breakdown for 17 regions, and the flows are valued in millions of won. The multiregional table breaks down for each region and for each product the intraregional elements, the values of transactions between each pair of regions, and the trade with the rest of the world on a product-by-product basis. For our purpose of assessing the efficiency of the different methodologies, including the FLQ method, the use of type B matrices is required.

Regarding the main closeness of fit measures proposed in the literature (Arto et al. 2014; Miller and Blair 2009; Tarancón 2002; Temurshoev et al. 2011; Valderas-Jaramillo et al. 2019) to assess the accuracy of the different methodologies, we use the Weighted Absolute Percentage Error (WAPE)[2] statistic, which is very frequently used in the input–output field. To define it formally, we call $x_{ij}$ the actual element $(i, j)$ of the matrix X $(m \times n)$ that we want to approximate using the regionalisation methodologies, and $\widetilde{x}_{ij}$ the estimated value for element $(i, j)$. Its expression is the following (Valderas-Jaramillo et al. 2019):

$$WAPE = \sum_{i=1}^{m} \sum_{j=1}^{n} \left( \frac{\left|x_{ij} - \widetilde{x}_{ij}\right|}{\sum_{i=1}^{m} \sum_{j=1}^{n} \left|x_{ij}\right|} \right). \tag{18}$$

The usefulness of this statistic comes from the elimination of the bias derived from giving the same weight to all the variables, since it measures the absolute error percentages, on average, weighted by the weight of each element. The values of other goodness-of-fit statistics confirming the results obtained from WAPE are presented in the Appendix A.

## 4. Results, Assessment, and Analysis

Initially, with the data from the multiregional table of the 17 regions of Korea in 2015, the results obtained from the estimation of the coefficients derived from the application of the FLQ, AFLQ, 2D-LQ, and ACILQ methodologies are contrasted with regard to the optimal values of the parameters in case the construction of the quotient requires it (Table 1).

As shown, in 82% of the Korean regions, a minimum value of the goodness-of-fit statistic is obtained from the use of the 2D-LQ methodology, and only in three regions the statistic is minimised using the FLQ methodology, and it is additionally found that the results obtained with the AFLQ and ACILQ methodologies do not improve compared with those obtained with the FLQ method.

In the case of the 2D-LQ ratio, all possible combinations of the parameters $\alpha$ and $\beta$ are tested based on Pereira-López et al. (2020). As mentioned above, given that this methodology has not been tested at the regional level, and in order to evaluate the sensitivity of each parameter, it was decided to give the $\alpha$ parameter values from 0 to 2, varying this parameter from 0.1 to 0.1, evaluating all possible combinations with the $\beta$ parameter. This last parameter receives all possible values from 0 to 1, varying in 0.01 increments.

The results of the values that minimise the statistic according to the 2D-LQ ratio are shown in Table 2.

**Table 1.** Values of the WAPE statistic with optimal parameters according to methodology.

| | Regional Size * (%) | FLQ | AFLQ | 2D-LQ | ACILQ | MINIMUM WAPE |
|---|---|---|---|---|---|---|
| 1. Gyeonggi-do | 22.85 | 41.1032 | 81.524 | 37.1578 | 57.7978 | 2D-LQ |
| 2. Seoul | 18.97 | 60.3231 | 976.6543 | 74.775 | 73.093 | FLQ |
| 3. Gyeongsangbuk-do | 7.00 | 55.7305 | 68.4707 | 45.3338 | 69.2217 | 2D-LQ |
| 4. Chungcheongnam-do | 6.96 | 69.4134 | 70.5892 | 58.7928 | 90.1735 | 2D-LQ |
| 5. Gyeongsangnam-do | 6.93 | 55.5216 | 71.6967 | 47.4954 | 61.2203 | 2D-LQ |
| 6. Ulsan | 6.32 | 74.3943 | 90.0028 | 57.863 | 93.3422 | 2D-LQ |
| 7. Incheon | 4.96 | 52.0759 | 56.8124 | 47.3581 | 86.4897 | 2D-LQ |
| 8. Jeollanam-do | 4.89 | 66.1587 | 90.7174 | 56.4066 | 75.8863 | 2D-LQ |
| 9. Busan | 4.73 | 50.2127 | 65.0186 | 43.8702 | 62.5571 | 2D-LQ |
| 10. Chungcheongbuk-do | 3.47 | 72.7577 | 79.2497 | 64.0712 | 92.4192 | 2D-LQ |
| 11. Jeollabuk-do | 2.82 | 63.2096 | 77.6713 | 56.5918 | 73.858 | 2D-LQ |
| 12. Daegu | 2.82 | 59.2067 | 103.1339 | 58.1531 | 70.0134 | 2D-LQ |
| 13. Gwangju | 2.07 | 69.6581 | 95.1578 | 56.8449 | 83.3072 | 2D-LQ |
| 14. Gangwon-do | 1.97 | 67.2204 | 72.4614 | 69.378 | 88.4054 | FLQ |
| 15. Daejeon | 1.92 | 80.4275 | 94.3827 | 71.1432 | 113.2094 | 2D-LQ |
| 16. Jeju-do | 0.81 | 70.6293 | 238.9962 | 72.3739 | 79.6553 | FLQ |
| 17. Sejong | 0.50 | 87.7021 | 93.8636 | 77.591 | 162.8571 | 2D-LQ |

* Share of gross output. Source: authors' calculations for Korea MRIO 2015. FLQ, AFLQ, 2D-LQ, and ACILQ are, respectively, the methodology FLQ, AFLQ, 2D-LQ, and ACILQ described in Section 2. WAPE is the Weighted Absolute Percentage Error statistic defined in Section 3.

**Table 2.** Optimal values of 2D-LQ alpha and beta parameters and WAPE statistic.

| Region | Regional Size * (%) | $\alpha$ | $\beta$ | WAPE |
|---|---|---|---|---|
| 1. Gyeonggi-do | 22.85 | 0 | 0.5 | 37.1578 |
| 2. Seoul | 18.97 | 2 | 0.52 | 74.7750 |
| 3. Gyeongsangbuk-do | 7.00 | 0.2 | 0.3 | 45.3338 |
| 4. Chungcheongnam-do | 6.96 | 0.1 | 0.37 | 58.7928 |
| 5. Gyeongsangnam-do | 6.93 | 0.2 | 0.26 | 47.4954 |
| 6. Ulsan | 6.32 | 0.1 | 0.27 | 57.8630 |
| 7. Incheon | 4.96 | 0.5 | 0.32 | 47.3581 |
| 8. Jeollanam-do | 4.89 | 0.1 | 0.26 | 56.4066 |
| 9. Busan | 4.73 | 0.7 | 0.23 | 43.8702 |
| 10. Chungcheongbuk-do | 3.47 | 0 | 0.38 | 64.0712 |
| 11. Jeollabuk-do | 2.82 | 0 | 0.26 | 56.5918 |
| 12. Daegu | 2.82 | 0 | 0.24 | 58.1531 |
| 13. Gwangju | 2.07 | 0.6 | 0.21 | 56.8449 |
| 14. Gangwon-do | 1.97 | 0 | 0.29 | 69.3780 |
| 15. Daejeon | 1.92 | 0 | 0.29 | 71.1432 |
| 16. Jeju-do | 0.81 | 1.8 | 0.3 | 72.3739 |
| 17. Sejong | 0.50 | 0 | 0.35 | 77.5910 |

* Share of gross output. Source: authors' calculations for Korea MRIO 2015.

It can be seen that for the province of Seoul that the value of the optimal $\alpha$ parameter is situated at the maximum possible value within the given range. For this reason, and exceptionally, it is decided to extend the range of values for this region and for this parameter. By allowing max $\alpha = 3$, a new optimum is reached in the pair $\alpha = 3$; $\beta = 0.52$, obtaining a value of the WAPE statistic = 74.7579, which represents an improvement of 0.02%, which is considered a nonsignificant improvement.

### 4.1. Proposal for Estimation of Parameter Values $\alpha$ and $\beta$

Similarly to what happens in the case of the $\delta$ parameter for the FLQ and AFLQ ratios, giving values for the parameters that modify the national coefficient in the case of 2D-LQ in regionalisation processes is problematic when no prior regional reference table is available.

The parameters α and β that smooth the rectification applied to the national coefficient matrix, according to the authors, are not associated with each other (López et al. 2013; Pereira-López et al. 2020), although both papers establish—in their practical application—ranges of combined optimality between values of α for a given β and, alternatively, a range of values of β for a given α.

Whether the superiority shown in the accuracy of the estimation of the 2D-LQ ratio (Table 1) can be considered generalisable or happens on an ad hoc basis needs to be assessed. Therefore, the construction of the 2D-LQ ratio should be reviewed in relation to the procedure established to obtain the values of the parameters α and β, incorporating an assignment of optimal values based on criteria established by economic theory. In this sense, this is a proposal for the estimation of the value of the parameter β which, combined with the range of values of the parameter α, provides a more accurate estimate of the values of the regional coefficients. Furthermore, given that there is greater sensitivity associated with changes in the β parameter compared with changes in the α parameter (Pereira-López et al. 2021), estimating the β parameter is considered crucial. From the information in Table 2, the combinations of α and β parameters that still maintain superiority in regard to the FLQ ratio were obtained (Table 3). As can be seen, the Chungcheongbuk-do region allows the combination of 769 different alternatives that offer a better estimate than FLQ, i.e., 36% of the possible combinations of the α and β parameters outperform the FLQ quotient. On average, for all regions, there are 19.4% of possible combinations of α and β parameters that are more accurate than the FLQ best estimate.

**Table 3.** Combinations of the parameters associated with the 2D-LQ ratio that offer better WAPE than optimal FLQ.

| Region | Regional Size * (%) | Number | (%) | α | | | β | | |
| --- | --- | --- | --- | --- | --- | --- | --- | --- | --- |
| | | | | Min | Max | Stand. Deviat. | Min | Max | Stand. Deviat. |
| 1. Gyeonggi-do | 22.85 | 579 | 27.3% | 0 | 2 | 0.606 | 0.35 | 0.66 | 0.081 |
| 3. Gyeongsangbuk-do | 7.00 | 197 | 9.3% | 0.1 | 0.7 | 0.185 | 0.16 | 0.5 | 0.089 |
| 4. Chungcheongnam-do | 6.96 | 334 | 15.8% | 0.1 | 0.9 | 0.236 | 0.19 | 0.64 | 0.119 |
| 5. Gyeongsangnam-do | 6.93 | 157 | 7.4% | 0.1 | 0.6 | 0.161 | 0.14 | 0.43 | 0.080 |
| 6. Ulsan | 6.32 | 231 | 10.9% | 0.1 | 0.6 | 0.159 | 0.11 | 0.58 | 0.121 |
| 7. Incheon | 4.96 | 144 | 6.8% | 0.1 | 0.8 | 0.207 | 0.23 | 0.46 | 0.060 |
| 8. Jeollanam-do | 4.89 | 617 | 29.1% | 0 | 2 | 0.606 | 0.15 | 0.49 | 0.086 |
| 9. Busan | 4.73 | 219 | 10.3% | 0.1 | 1.4 | 0.338 | 0.15 | 0.35 | 0.054 |
| 10. Chungcheongbuk-do | 3.47 | 769 | 36.3% | 0 | 2 | 0.605 | 0.23 | 0.62 | 0.106 |
| 11. Jeollabuk-do | 2.82 | 498 | 23.5% | 0 | 2 | 0.605 | 0.18 | 0.44 | 0.069 |
| 12. Daegu | 2.82 | 167 | 7.9% | 0 | 2 | 0.604 | 0.21 | 0.3 | 0.024 |
| 13. Gwangju | 2.07 | 381 | 18.0% | 0.1 | 1.5 | 0.382 | 0.11 | 0.42 | 0.082 |
| 15. Daejeon | 1.92 | 764 | 36.0% | 0 | 2 | 0.606 | 0.18 | 0.62 | 0.108 |
| 17. Sejong | 0.50 | 700 | 33.0% | 0 | 2 | 0.606 | 0.26 | 0.72 | 0.106 |

* Share of gross output. Source: authors' calculations for Korea MRIO 2015.

Regarding the range of values that the parameters can take, it is observed that the range of possible values of β is smaller than that of α. Thus, for all the regions, on average, the standard deviation of all the possible values of the α parameter is 0.422, while for the β parameter it is 0.085.

Table 4 shows the range of possible values of parameter α by selecting the value of parameter β from the most accurate combination of the two.

**Table 4.** Range of values of the parameter α fixed to the parameter β that minimises WAPE, for which 2D-LQ is more accurate than the FLQ quotient.

| Region | Regional Size * (%) | Number | (%) | α | | |
|---|---|---|---|---|---|---|
| | | | | Min | Max | Standard Deviation |
| 1. Gyeonggi-do | 22.85 | 21 | 100.00% | 0 | 2 | 0.620 |
| 3. Gyeongsangbuk-do | 7.00 | 7 | 33.30% | 0.1 | 0.7 | 0.216 |
| 4. Chungcheongnam-do | 6.96 | 8 | 38.10% | 0.1 | 0.8 | 0.245 |
| 5. Gyeongsangnam-do | 6.93 | 6 | 28.60% | 0.1 | 0.6 | 0.187 |
| 6. Ulsan | 6.32 | 5 | 23.80% | 0.1 | 0.5 | 0.158 |
| 7. Incheon | 4.96 | 7 | 33.30% | 0.1 | 0.7 | 0.216 |
| 8. Jeollanam-do | 4.89 | 21 | 100.00% | 0 | 2 | 0.620 |
| 9. Busan | 4.73 | 12 | 57.10% | 0.1 | 1.2 | 0.361 |
| 10. Chungcheongbuk-do | 3.47 | 21 | 100.00% | 0 | 2 | 0.620 |
| 11. Jeollabuk-do | 2.82 | 21 | 100.00% | 0 | 2 | 0.620 |
| 12. Daegu | 2.82 | 21 | 100.00% | 0 | 2 | 0.620 |
| 13. Gwangju | 2.07 | 15 | 71.40% | 0.1 | 1.5 | 0.447 |
| 15. Daejeon | 1.92 | 21 | 100.00% | 0 | 2 | 0.620 |
| 17. Sejong | 0.50 | 14 | 66.70% | 0 | 1.3 | 0.418 |

* Share of gross output. Source: authors' calculations for Korea MRIO 2015.

As shown, in 36% of the regions, fixing the value of the parameter β to any value given to the parameter α guarantees a better estimate than that obtained by the FLQ ratio. In this case, the average of the standard deviations for all regions is 0.427.

This result justifies that, in the process of finding which values to give to the parameters associated with the 2D-LQ ratio, it is recommended, in the first place, to select an appropriate value for $\beta$. At this point, it is worth remembering that the parameters $\alpha$ and $\beta$ are associated with the degree of rectification applied to the rows and columns of the matrix to incorporate the existence of interregional trade. Therefore, for the estimation of the parameter $\beta$ applied to the Korean regions in 2015, we propose a regression equation in which the explanatory variables are road freight transport (origin and destination) and regional size. Formally, the regression equation is:

$$\hat{\beta} = 1.64RS + 0.83FIT + \text{e.} \tag{19}$$

where *RS* represents the regional size measured in terms of gross output, *FIT* corresponds to the weight of freight transport flow from other regions (interregional transport) measured in tonnes over the total freight transport flow, including both the interregional transport flow and the transport flow generated within the region itself (intraregional), also measured in tonnes, and e is the residual. The two regressors are statistically significant at 1%, and the model has a value of $R^2 = 0.837$.

Similarly, for the estimation of the parameter $\alpha$, the following equation is proposed:

$$\hat{\alpha} = 1.66RS + 0.82FET + \text{e.} \tag{20}$$

In this case, the regressors are also statistically significant at 1% and with an $R^2 = 0.838$, where *RS* represents the regional size measured in terms of gross output, *FET* corresponds to the weight of the transport flow of goods destined for other regions (interregional transport) measured in tonnes over the total transport flow of goods, including both the interregional transport flow and the transport flow generated within the region itself (intra-regional), also measured in tonnes, and e is the residual.

The results of the application of Equations (19) and (20) are shown in Table 5, in which the estimated values of the parameters $\hat{\alpha}$ and $\hat{\beta}$ and the values corresponding to the WAPE statistic are shown. As can be seen, the results are conclusive in that 79% of the cases the estimate obtained is still higher than the best estimate obtained from the FLQ

ratios (omitting the regions of Seoul, Gangwon-do, and Jeju-do, where FLQ is superior in precision). In the case of Gwangju, the difference of the estimate in regard to the best WAPE obtained by FLQ is 7.8%, in Daegu this difference increases slightly to 8.8%, and in the case of Daejeon, the difference rises to 25%.

**Table 5.** Value of the parameters $\hat{\alpha}$ and $\hat{\beta}$ estimated and value of the WAPE statistic.

| Region | Regional Size * (%) | $\hat{\alpha}$ | $\hat{\beta}$ | WAPE |
|---|---|---|---|---|
| 1. Gyeonggi-do | 22.85 | 0.5881 | 0.5928 | 39.0869 |
| 2. Seoul | 18.97 | 0.4712 | 0.4661 | 75.1024 |
| 3. Gyeongsangbuk-do | 7.00 | 0.2837 | 0.2787 | 46.1892 |
| 4. Chungcheongnam-do | 6.96 | 0.3601 | 0.3543 | 60.7581 |
| 5. Gyeongsangnam-do | 6.93 | 0.3251 | 0.3227 | 50.1645 |
| 6. Ulsan | 6.32 | 0.2728 | 0.2821 | 60.3819 |
| 7. Incheon | 4.96 | 0.3336 | 0.3335 | 48.4683 |
| 8. Jeollanam-do | 4.89 | 0.2292 | 0.2236 | 57.4244 |
| 9. Busan | 4.73 | 0.2557 | 0.2635 | 47.6777 |
| 10. Chungcheongbuk-do | 3.47 | 0.3176 | 0.3247 | 65.2028 |
| 11. Jeollabuk-do | 2.82 | 0.2037 | 0.1973 | 60.1445 |
| 12. Daegu | 2.82 | 0.1451 | 0.1522 | 64.4493 |
| 13. Gwangju | 2.07 | 0.1256 | 0.1263 | 75.1186 |
| 14. Gangwon-do | 1.97 | 0.2480 | 0.2400 | 72.4061 |
| 15. Daejeon | 1.92 | 0.1194 | 0.1209 | 100.5671 |
| 16. Jeju-do | 0.81 | 0.0135 | 0.0133 | 151.0019 |
| 17. Sejong | 0.50 | 0.4475 | 0.4469 | 79.2421 |

* Share of gross output. Source: authors' calculations for Korea MRIO 2015.

On average, 2D-LQ obtains an estimation accuracy gain of 10% with our proposal.

### 4.2. Application to the Case of Spanish Regions

To test and evaluate the validity of the estimating Equations (19) and (20) proposed for Korean regions in other contexts, we use the case of Spanish regions to estimate the values of $\alpha$ and $\beta$.

Spain does not have a multiregional table, and the availability of regional tables is neither standardised nor homogenised across regions. However, taking into account that regional sizes in terms of gross output have remained practically constant from 2005 to the present, and that the pattern of trade must not have changed substantially in the territories, we proceed to homogenise the largest number of available regional tables in relation to the national tables taken as a reference between 2005 and 2015. For the estimation of the optimal values of the parameters, in the case of Spain, the largest possible number of rows/columns is maintained independently in each of them instead of homogenising all the regional tables to the same number of rows and columns. The reason for this is that excessive aggregation into branches can distort the regionalisation process (Flegg et al. 2014) and may lead to erroneous conclusions, as can be seen in (Riddington et al. 2006).

Of the seventeen existing Spanish regions, thirteen of them have an input–output table available, which allows for the application of the procedure in this context.

First, for these regions, the different methods analysed are compared using, as for the Korean regions, the WAPE statistic.

Table 6 presents the results obtained for the different Spanish regional input–output Tables, indicating the reference year. In the case of the 2D-LQ quotient, we proceed by checking all possible combinations of the parameters $\alpha$ and $\beta$ by giving the parameter $\alpha$ values from 0 to 2, varying this parameter from 0.1 to 0.1, and evaluating all possible combinations with the parameter $\beta$. The parameter $\beta$ takes all possible values from 0 to 1 in 0.01 increments. It is observed that the cases of Andalusia and the Balearic Islands constitute an exception in the evaluation, insofar as the minimum values of the statistic are obtained with the maximum possible value of $\alpha$, namely $\alpha = 2$, so the range of values for

these two cases is extended, taking values from 0 to 3. Andalusia obtains the minimum WAPE, with an $\alpha$ = 2.7 while, in the case of the Balearic Islands, the minimum WAPE is obtained with $\alpha$ = 3.

**Table 6.** Closeness of fit, according to WAPE, between different types of location quotients.

| Region/Year | Regional Size * (%) | WAPE | | | | | |
|---|---|---|---|---|---|---|---|
| | | FLQ | AFLQ | 2D-LQ | ACILQ | MINIMUM | |
| Catalonia 2011 | 20.53 | 93.328 | 129.356 | 108.862 | 98.172 | 93.328 | FLQ |
| Community of Madrid 2010 | 18.87 | 77.905 | 83.981 | 76.910 | 89.381 | 76.910 | 2D-LQ |
| Andalusia 2010 | 13.20 | 60.655 | 66.761 | 55.726 | 62.871 | 55.726 | 2D-LQ |
| Basque Country 2015 | 6.62 | 63.174 | 72.237 | 63.198 | 67.947 | 63.174 | FLQ |
| Galicia 2011 | 5.44 | 69.420 | 73.013 | 63.198 | 71.180 | 63.198 | 2D-LQ |
| Canary Islands 2005 | 3.50 | 84.930 | 92.687 | 76.539 | 89.014 | 76.539 | 2D-LQ |
| Castilla-La Mancha 2005 | 3.48 | 73.090 | 81.874 | 69.252 | 75.958 | 69.252 | 2D-LQ |
| Aragon 2005 | 3.25 | 88.096 | 96.102 | 92.194 | 123.387 | 88.096 | FLQ |
| Balearic Islands 2004 | 2.19 | 75.940 | 91.000 | 73.688 | 75.927 | 73.688 | 2D-LQ |
| Principality of Asturias 2015 | 1.89 | 77.158 | 90.228 | 73.826 | 78.785 | 73.826 | 2D-LQ |
| Community of Navarra 2010 | 1.88 | 73.724 | 86.877 | 68.909 | 75.494 | 68.909 | 2D-LQ |
| Cantabria 2015 | 1.09 | 74.092 | 79.468 | 69.459 | 77.478 | 69.459 | 2D-LQ |
| La Rioja 2008 | 0.77 | 83.874 | 86.426 | 80.089 | 97.820 | 80.089 | 2D-LQ |

* Share of gross output. Source: authors' calculations for Spain regional IOTs.

As can be seen, the results indicate that it is the 2D-LQ method that obtains greater precision in most Spanish regions. However, the ACILQ ratio does not improve, in any case, the rest of the ratios, so it can be stated that the smoothing performed on the CILQ ratio is not enough to improve the estimation when compared with the rest of the ratios.

Applying again the proposed procedure for the estimation of the parameters $\alpha$ and $\beta$, the following results are obtained:

$$\hat{\beta} = 1.78RS + 0.47FIT + \text{e}. \tag{21}$$

where, as noted above, *RS* represents the regional size measured in terms of gross output, *FIT* corresponds to the weight of freight transport flow from other regions (interregional transport) measured in tonnes over the total freight transport flow, and e is the residual. The two regressors are statistically significant at 1%, and the regression equation has a value of $R^2$ = 0.704.

In the case of the parameter estimating equation (smoothing the row rectification from the simple location quotient SLQ), the best specification is achieved in logarithmic terms:

$$ln\hat{\alpha} = 0.5681lnRE - 0.4228lnFET + e. \tag{22}$$

where, now, the *RE* variable represents the relative regional size measured in terms of employment, while the *FET* variable corresponds to the weight of the transport flow of goods destined for other regions (interregional transport) measured in tonnes over the total transport flow of goods, including both the interregional transport flow and the transport flow generated within the same region (intraregional), also measured in tonnes, and e is the residual. The variables RE and FET are statistically significant at 1% and 5%, respectively, and the model has a value of $R^2$ = 0.572.

Table 7 presents the estimated $\alpha$ and $\beta$ values, the WAPE statistic, and the relative difference with regard to the optimal WAPE.

**Table 7.** Value of the estimated parameters $\hat{\alpha}$ and $\hat{\beta}$ and value of the WAPE statistic.

| Region/Year | Regional Size * (%) | $\hat{\alpha}$ | $\hat{\beta}$ | WAPE | Dev. s/Optimum |
|---|---|---|---|---|---|
| 1. Catalonia 2011 | 20.53 | 0.7939 | 0.4115 | 126.6419 | 35.70% |
| 2. Community of Madrid 2010 | 18.87 | 0.5362 | 0.5192 | 77.1477 | 0.31% |
| 3. Andalusia 2010 | 13.20 | 0.8200 | 0.3076 | 56.8420 | 2.00% |
| 4. Basque Country 2015 | 6.62 | 0.3001 | 0.3568 | 65.1524 | 3.13% |
| 5. Galicia 2011 | 5.44 | 0.4485 | 0.3010 | 68.0631 | 7.70% |
| 6. Canary Islands 2005 | 3.50 | 11.505 | 0.0683 | 93.8604 | 22.63% |
| 7. Castilla-La Mancha 2005 | 3.48 | 0.2264 | 0.2395 | 69.5357 | 0.41% |
| 8. Aragon 2005 | 3.25 | 0.2145 | 0.2264 | 108.1597 | 22.78% |
| 9. Balearic Islands 2004 | 2.19 | 12.118 | 0.0453 | 84.0523 | 14.06% |
| 10. Principality of Asturias 2015 | 1.89 | 0.1933 | 0.1354 | 74.7481 | 1.25% |
| 11. Community of Navarra 2010 | 1.88 | 0.1333 | 0.0950 | 72.4648 | 5.16% |
| 12. Cantabria 2015 | 1.09 | 0.1217 | 0.1761 | 69.9141 | 0.66% |
| 13. La Rioja 2008 | 0.77 | 0.0784 | 0.1859 | 82.2648 | 2.72% |

* Share of gross output. Source: authors' calculations for Spain regional IOTs.

As shown, the values of the parameters α and β obtained with the proposed estimation (21) and (22) obtain values of the fit statistic whose deviation from the optimum is quite acceptable and, in general, better values of the statistic are still obtained than the rest of the methodologies based on location quotients. The largest differences in terms of fit to the optimum are found in the case of the Spanish regions that offer the best estimates with the FLQ methodology.

Once again, the superiority of the 2D-LQ methodology is evident when using the regression procedure proposed in this paper to estimate the values of the parameters α and β. Thus, in 61.5% of the cases, the modified 2D-LQ methodology outperforms in closeness of fit the best possible estimate obtained from the FLQ methodology using the optimal value of δ.

## 5. Discussion and Conclusions

This paper evaluates the results obtained from regionalisation methodologies based on location quotients: FLQ, AFLQ, 2D-LQ, and ACILQ, with the aim of choosing the best technique, especially for contexts where previous input–output frameworks do not exist and statistical information is rather limited. Specifically, in a novel way, a comparison is made at the regional level (for the Korean regions) of the goodness-of-fit of the two best performing methodologies, FLQ and 2D-LQ. Having verified the goodness of fit, as well as the lower sensitivity in the variation of the parameters that smooth the correction of the national coefficients of the parameters associated with the 2D-LQ ratio, this work concludes that the 2D-LQ alternative is the one that obtains the most accurate results in 14 of the 17 Korean regions. Therefore, the 2D-LQ method improves the estimates of the FLQ technique and is considered a useful technique in the estimation of regional input–output tables. Having verified the superiority of the 2D-LQ method, this paper reviews and modifies the procedure for obtaining the values of the parameters α and β, incorporating an assignment of optimal values based on criteria established by economic theory. To this end, for the first time, a novel proposal is made for the estimation of the unknown parameters α and β from a regression equation that uses information on regional size and road freight transport as explanatory variables.

The superiority at the regional level of the 2D-LQ method over other FLQ methodologies confirms the results previously obtained at the national level (Pereira-López et al. 2020, 2021).

The results obtained are conclusive and unequivocal in that 79% of the cases the estimate obtained is still higher than the best estimate obtained from the FLQ ratios. This finding can be considered relevant for the estimation of regional input–output tables in contexts where there is no multiregional input–output framework, as in the case of the

Korean regions, which have multiregional input–output survey tables. Moreover, in the case of the Spanish regions (in this case without an integrated multiregional framework), the superiority of the 2D-LQ method is verified (10 of the 13 Spanish regions), as well as the validity of the proposal for estimating the parameters α and β, since in 61.5% of the cases more precise estimates are still obtained than with the FLQ method.

The aforementioned results support a first recommendation to estimate the domestic intermediate input matrix using the 2D-LQ bidimensional location quotient technique in contexts where no regional input–output framework exists.

Our second recommendation for the determination of the parameters α and β in the 2D-LQ bidimensional location quotient method is to use the regression equation of our proposal, in which the regressors are the regional size measured in terms of Gross Domestic Product, GDP, and the weight of freight transport flow from other regions (interregional transport) over the total freight transport flow. The values of the parameters obtained with the proposed estimation provide values of the fit statistic whose deviation from the optimum is minimal.

The third recommendation is to confirm that input–output tables provide a useful tool for analysing economic and environmental impacts. The availability of the intermediate demand matrix allows for the extraction of income multipliers and employment multipliers to assess the economic contribution of different economic activities, for instance, in the economic analysis of the tourism industry. This can intuitively be expected to play an important role in economic growth and employment in many regions, however, using the input–output model, the relative high potential of tourism can be measured more accurately compared with other sectors.

Finally, future research could test other methodological procedures that improve upon this proposal in terms of goodness-of-fit. In this regard, accuracy assessment could be extended to the Demand- and Supply-based Location Quotient method (Fujimoto 2019) or hybrid procedures based on the use of augmented location quotients (Jahn 2017).

**Author Contributions:** Conceptualization, R.M.-A. and J.D.B.-A.; methodology, R.M.-A. and J.D.B.-A.; software, R.M.-A.; validation, R.M.-A., J.D.B.-A. and, M.d.M.S.-d.-l.-V.; formal analysis, R.M.-A., J.D.B.-A., and M.d.M.S.-d.-l.-V.; investigation, R.M.-A., J.D.B.-A. and M.d.M.S.-d.-l.-V. All authors have read and agreed to the published version of the manuscript.

**Funding:** This research received no external funding.

**Informed Consent Statement:** Not applicable.

**Data Availability Statement:** Data is sourced from the Bank of Korea website (https://ecos.bok.or.kr/).

**Acknowledgments:** The authors would like to thank the anonymous referees, the academic editor, and the journal's assistant editor for their fruitful comments and suggestions to increase the quality of this manuscript.

**Conflicts of Interest:** The authors declare no conflict of interest.

## Appendix A

The results obtained from the Weighted Absolute Scaled Error, WASE, Symmetric Mean Absolute Percent Error, SWAPE, $\rho$-SWAPE, and Similarity Index, IS, goodness-of-fit statistics are defined and presented below. Following the nomenclature used above, where

$x_{ij}$ is the actual element $(i,j)$ of the survey matrix X $(m \times n)$ and $\widetilde{x}_{ij}$ the estimated value for element $(i,j)$, we define the following statistics, Valderas-Jaramillo et al. (2019):

$$\rho - SWAPE = 100\left(1 - \frac{SWAPE}{200}\right).$$

$$\text{Where } SWAPE = 200 \sum_{i=1}^{m} \sum_{j=1}^{n} \left(\frac{|x_{ij}|}{\Sigma_{i=1}^{m}\Sigma_{j=1}^{n}|x_{ij}|}\right)\left|\frac{x_{ij} - \widetilde{x}_{ij}}{x_{ij} + \widetilde{x}_{ij}}\right|.$$

$$WASE = \sum_{i=1}^{m} \sum_{j=1}^{n} \left(\frac{|x_{ij}|}{\Sigma_{i=1}^{m}\Sigma_{j=1}^{n}|x_{ij}|}\right)\left|\frac{|x_{ij} - \widetilde{x}_{ij}|}{\frac{\Sigma_{i=1}^{m}\Sigma_{j=1}^{n}|x_{ij} - \overline{x}_{ij}|}{mn}}\right|.$$

$$IS = 50\left(1 + r_{X,\overline{X}}\right).$$

$$\text{where } r_{X,\overline{X}} = \frac{Cov\left(X,\overline{X}\right)}{S_X S_{\overline{X}}}.$$

The Weighted Absolute Scaled Error (WASE) statistic offers a lower sensitivity to anomalous elements as it is not affected by changes in scale, origin or size of coefficients (Valderas 2015). Based on the work developed by Arto et al. (2014), the ρ-SWAPE is presented, whose interpretation is similar to a coefficient of determination, taking a unit value if a perfect fit is obtained and zero otherwise. It is a suitable method for comparing different methods (Valderas 2015). Finally, the Similarity Index (SI) is presented, which shows a more perfect fit the closer the value of the index is to 100 (Valderas 2015).

**Table A1.** Values of the complementary statistics from the estimation of the methodologies with the parameter values minimising the WAPE statistic.

| | FLQ | | | | |
|---|---|---|---|---|---|
| **NAME** | **DELTA_FLQ** | **WAPE** | **ρ-SWAPE** | **WASE** | **IS** |
| 1. Gyeonggi-do | 0.545 | 41.1032 | 0.8938 | 4.0671 | 93.0768 |
| 2. Seoul | 0.186 | 60.3231 | 0.8683 | 2.2427 | 92.7361 |
| 3. Gyeongsangbuk-do | 0.353 | 55.7305 | 0.8606 | 4.3549 | 91.5913 |
| 4. Chungcheongnam-do | 0.642 | 69.4134 | 0.7918 | 6.0804 | 85.0194 |
| 5. Gyeongsangnam-do | 0.293 | 55.5216 | 0.8617 | 4.1643 | 91.2083 |
| 6. Ulsan | 0.594 | 74.3943 | 0.7915 | 6.0399 | 77.363 |
| 7. Incheon | 0.469 | 52.0759 | 0.8587 | 5.7447 | 86.6191 |
| 8. Jeollanam-do | 0.288 | 66.1587 | 0.8451 | 6.0977 | 90.2414 |
| 9. Busan | 0.281 | 50.2127 | 0.8747 | 3.8122 | 92.1465 |
| 10. Chungcheongbuk-do | 0.487 | 72.7577 | 0.7842 | 6.5047 | 87.3664 |
| 11. Jeollabuk-do | 0.335 | 63.2096 | 0.825 | 5.9123 | 89.5707 |
| 12. Daegu | 0.26 | 59.2067 | 0.8503 | 6.1472 | 89.8076 |
| 13. Gwangju | 0.345 | 69.6581 | 0.8096 | 7.6022 | 84.0692 |
| 14. Gangwon-do | 0.284 | 67.2204 | 0.8177 | 10.323 | 88.5964 |
| 15. Daejeon | 0.451 | 80.4275 | 0.7715 | 12.2237 | 76.8347 |
| 16. Jeju-do | 0.208 | 70.6293 | 0.8275 | 7.3354 | 90.0979 |
| 17. Sejong | 0.605 | 87.7021 | 0.6724 | 37.1736 | 68.1598 |

**Table A1.** *Cont.*

| 2D-LQ | | | | | | |
|---|---|---|---|---|---|---|
| **NAME** | **ALPHA** | **BETA** | **WAPE** | **ρ-SWAPE** | **WASE** | **IS** |
| 1. Gyeonggi-do | 0 | 0.5 | 37.1578 | 0.9014 | 3.9688 | 93.9211 |
| 2. Seoul | 2 | 0.52 | 74.775 | 0.7867 | 4.2657 | 88.0443 |
| 3. Gyeongsangbuk-do | 0.2 | 0.3 | 45.3338 | 0.8793 | 5.258 | 91.5255 |
| 4. Chungcheongnam-do | 0.1 | 0.37 | 58.7928 | 0.8319 | 5.5183 | 90.0182 |
| 5. Gyeongsangnam-do | 0.2 | 0.26 | 47.4954 | 0.8785 | 4.6861 | 90.935 |
| 6. Ulsan | 0.1 | 0.27 | 57.863 | 0.8429 | 5.3416 | 82.5661 |
| 7. Incheon | 0.5 | 0.32 | 47.3581 | 0.8757 | 5.1793 | 90.0882 |
| 8. Jeollanam-do | 0.1 | 0.26 | 56.4066 | 0.851 | 6.9385 | 92.8184 |
| 9. Busan | 0.7 | 0.23 | 43.8702 | 0.8882 | 3.7099 | 93.0696 |
| 10. Chungcheongbuk-do | 0 | 0.38 | 64.0712 | 0.8116 | 7.2195 | 88.1739 |
| 11. Jeollabuk-do | 0 | 0.26 | 56.5918 | 0.8504 | 6.0231 | 91.9699 |
| 12. Daegu | 0 | 0.24 | 58.1531 | 0.8486 | 6.1514 | 89.3814 |
| 13. Gwangju | 0.6 | 0.21 | 56.8449 | 0.8517 | 6.8713 | 87.796 |
| 14. Gangwon-do | 0 | 0.29 | 69.378 | 0.8042 | 11.0414 | 89.4621 |
| 15. Daejeon | 0 | 0.29 | 71.1432 | 0.8072 | 10.4659 | 86.4103 |
| 16. Jeju-do | 1.8 | 0.3 | 72.3739 | 0.7955 | 9.1072 | 90.0007 |
| 17. Sejong | 0 | 0.35 | 77.591 | 0.7506 | 33.0359 | 87.6514 |

WAPE, ρ-SWAPE, WASE and IS are, respectively, the Weighted Absolute Percentage Error, ρ-Symmetric Mean Absolute Percent Error, Weighted Absolute Scaled Error and Similarity Index goodness-of-fit statistics defined and presented above.

**Table A2.** Values of the complementary statistics from the proposed estimation of the parameters of the 2D-LQ methodology.

| **NAME** | **ALPHA** | **BETA** | **WAPE** | **ρ-SWAPE** | **WASE** | **IS** |
|---|---|---|---|---|---|---|
| 1. Gyeonggi-do | 0.5881 | 0.5928 | 39.0869 | 0.8907 | 4.3296 | 93.6326 |
| 2. Seoul | 0.4712 | 0.4661 | 75.1024 | 0.7932 | 4.0248 | 88.2707 |
| 3. Gyeongsangbuk-do | 0.2837 | 0.2787 | 46.1892 | 0.8799 | 5.151 | 91.1619 |
| 4. Chungcheongnam-do | 0.3601 | 0.3543 | 60.7581 | 0.8288 | 5.4471 | 89.0829 |
| 5. Gyeongsangnam-do | 0.3251 | 0.3227 | 50.1645 | 0.861 | 5.3012 | 89.3275 |
| 6. Ulsan | 0.2728 | 0.2821 | 60.3819 | 0.8288 | 5.6868 | 80.2831 |
| 7. Incheon | 0.3336 | 0.3335 | 48.4683 | 0.8692 | 5.3418 | 90.6737 |
| 8. Jeollanam-do | 0.2292 | 0.2236 | 57.4244 | 0.8578 | 6.3423 | 92.7212 |
| 9. Busan | 0.2557 | 0.2635 | 47.6777 | 0.8717 | 4.1725 | 92.475 |
| 10. Chungcheongbuk-do | 0.3176 | 0.3247 | 65.2028 | 0.824 | 6.6471 | 88.5009 |
| 11. Jeollabuk-do | 0.2037 | 0.1973 | 60.1445 | 0.8585 | 5.1246 | 91.9652 |
| 12. Daegu | 0.1451 | 0.1522 | 64.4493 | 0.8582 | 4.9516 | 89.3686 |
| 13. Gwangju | 0.1256 | 0.1263 | 75.1186 | 0.8431 | 5.259 | 89.4027 |
| 14. Gangwon-do | 0.248 | 0.24 | 72.4061 | 0.821 | 9.7565 | 89.1708 |
| 15. Daejeon | 0.1194 | 0.1209 | 100.567 | 0.8146 | 7.1664 | 86.3329 |
| 16. Jeju-do | 0.0135 | 0.0133 | 151.002 | 0.7765 | 4.1075 | 86.9295 |
| 17. Sejong | 0.4475 | 0.4469 | 79.2421 | 0.7168 | 34.4763 | 87.8191 |

WAPE, ρ-SWAPE, WASE and IS are, respectively, the Weighted Absolute Percentage Error, ρ-Symmetric Mean Absolute Percent Error, Weighted Absolute Scaled Error and Similarity Index goodness-of-fit statistics defined and presented above.

## Note

[1]  Bakhtiari and Dehghanizadeh (2012) offer an alternative, called the adjusted interindustry location quotient (ACILQ), which consists of adjusting the CILQ quotient based on the size of the region whose table is to be estimated. $ACILQ = CILQ * K$, where $K = \frac{e^m - e^{-m}}{e^m + e^{-m}}$ y $m = 10 \left[ \frac{x^r}{x^n} \right]$. Due to the tangential structure of the adjustment parameter *K*, it is guaranteed that this parameter takes values in the interval [0,1]. Although it is true that in the case study offered for the province of Yazd, in Iran, it improves the SLQ, CILQ, and FLQ ratios for all sectors, it is no less true that the methodology of comparing results based on absolute deviations from the mean or from the mean, without weighting, is not the one normally used in the discipline, so that this

methodology, while certainly attractive, should be subjected to evaluation in other contexts, territories, and according to the commonly used goodness-of-fit statistics.

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
