# Peer review of "A New Improvement Proposal to Estimate Regional Input–Output Structure Using the 2D-LQ Approach"

_economies, doi:10.3390/economies11010020_

Round 1

Reviewer 1 Report

An article is well organized and well done. An issue is deeply explored. But, in my concern, a little bit more additional disclosure required, for though individuals, who a not close to mathematics

Author Response

We sincerely appreciate your comments and suggestions, which have indeed improved the work. Following your indication, we have added some additional disclosure.

Reviewer 2 Report

I appreciate the article and the difficult question of estimating parameters when I-O tables are unavailable.  The article is original. Here is some advice. 

ABSTRACT:

I suggest better explaining the motivation of the study to offer a more appealing for the readers. Please start with the research needs, the main research questions, and the main findings.

METHODOLOGY:

It is well-described and helpful in explaining the research question. Also, the appendix is useful to specify some methodology steps. 

I suggest clarifying why you chose those two case studies to check the methodology. 

DISCUSSION: 

The discussions about the findings could be improved. 

I suggest better connecting the results with the findings and research question. You can also specify the important contribution of the tourism economics analysis. 

Author Response

Answer

We sincerely appreciate your comments and suggestions, which have indeed improved the work. Here we include the changes made to the manuscript after each of your comments.

ABSTRACT:

I suggest better explaining the motivation of the study to offer a more appealing for the readers. Please start with the research needs, the main research questions, and the main findings.

Answer:

Done.

METHODOLOGY:

It is well-described and helpful in explaining the research question. Also, the appendix is useful to specify some methodology steps. 

I suggest clarifying why you chose those two case studies to check the methodology. 

Answer:

Following your indication, we have included the following:

“Korea's multi-regional table for 2015 has been used for the empirical study, as it is a recent database with a wide availability of survey data at the regional level. However, in the case of regions without a multi-regional input-output framework the application has been done in the Spanish regions.”

DISCUSSION: 

The discussions about the findings could be improved. 

I suggest better connecting the results with the findings and research question. You can also specify the important contribution of the tourism economics analysis. 

 Answer:

Following your suggestions, we have incorporated this information (p.16 and 17), as set out below.

The aforementioned results support a first recommendation to estimate the domestic intermediate input matrix using the 2D-LQ bi-dimensional location quotient technique in contexts where no regional input-output framework exists.

Our second recommendation for the determination of the parameters α and β in the 2DLQ method is to use the regression equation of our proposal in which the regressors are the regional size measured in terms of GDP and the weight of freight transport flow from other regions (inter-regional transport) over the total freight transport flow. The values of the parameters obtained with the proposed estimation provide values of the fit statistic whose deviation from the optimum is minimal.

The third recommendation is to confirm that input–output tables provide a useful tool for analysing economic and environmental impacts. The availability of the intermediate demand matrix allows for the extraction of income multipliers and employment multipliers to assess the economic contribution of different economic activities, for instance in the economic analysis of the tourism industry. This can intuitively be expected to play an important role in economic growth and employment in many regions, however, using the input-output model, the relative high potential of tourism can be measured more accurately compared to other sectors.

Finally, future research could test other methodological procedures that improve upon this proposal in terms of goodness-of-fit. In this regard, accuracy assessment could be extended to the Demand and Suply based Location Quotient method (Fujimoto, 2019) or hybrid procedures based on the use of augmented location quotients (Jahn, 2017).

Reviewer 3 Report

1.       Abstract: "This paper evaluates the goodness-of-fit of regionalization methodologies based on location quotients for the estimation of input-output table " why are you interested in this evaluates what issue of the current research?

2.       Abstract : What is the numerical result of the proposed model/methods compared to the present issue addressing a similar or related problem?

3.       Introduction : Clearly state research purpose, research gap, and research contribution I cannot locate these three concerns inside the introduction.

4.       What is the objective to address? “Methodological review on location quotients” This part should describe the approach used to identify the answer to the research gap, the research contribution, and the means through which these objectives can be attained. This section must be rewritten so that others can follow in the author's footsteps and attain the same goal.

5.       Discussion section : It should inform the reader how your findings are comparable or distinct to, or support or do not support, other studies. In addition, it should address the academic and corporate or social implications of your findings. Your only discussion is the conclusion. No academic or societal implications are demonstrated, and no other relevant literature is presented.

6.       Conclusion, recommendation, limitation and future research is totally missing.

Round 2

Reviewer 3 Report

Well done